# Proteasome Inhibitors and Their Pharmacokinetics, Pharmacodynamics, and Metabolism

**DOI:** 10.3390/ijms222111595

**Published:** 2021-10-27

**Authors:** Jinhai Wang, Ying Fang, R. Andrea Fan, Christopher J. Kirk

**Affiliations:** Kezar Life Sciences, South San Francisco, CA 94080, USA; yfang@kezarbio.com (Y.F.); afan@kezarbio.com (R.A.F.); ckirk@kezarbio.com (C.J.K.)

**Keywords:** proteasome, proteasome inhibitors, immunoproteasome, immunoproteasome inhibitors, epoxide hydrolases, microsomal epoxide hydrolase

## Abstract

The proteasome is responsible for mediating intracellular protein degradation and regulating cellular function with impact on tumor and immune effector cell biology. The proteasome is found predominantly in two forms, the constitutive proteasome and the immunoproteasome. It has been validated as a therapeutic drug target through regulatory approval with 2 distinct chemical classes of small molecular inhibitors (boronic acid derivatives and peptide epoxyketones), including 3 compounds, bortezomib (VELCADE), carfilzomib (KYPROLIS), and ixazomib (NINLARO), for use in the treatment of the plasma cell neoplasm, multiple myeloma. Additionally, a selective inhibitor of immunoproteasome (KZR-616) is being developed for the treatment of autoimmune diseases. Here, we compare and contrast the pharmacokinetics (PK), pharmacodynamics (PD), and metabolism of these 2 classes of compounds in preclinical models and clinical studies. The distinct metabolism of peptide epoxyketones, which is primarily mediated by microsomal epoxide hydrolase, is highlighted and postulated as a favorable property for the development of this class of compound in chronic conditions.

## 1. Introduction

The ubiquitin/proteasome system constitutes one of the primary means by which intracellular proteins are degraded. Ubiquitin/proteasome-regulated pathways contribute to the dynamic control of key cell signaling components and the maintenance of overall cellular homeostasis, as well as are associated with multiple pathological conditions, including cancer, autoimmune disorders, and neurodegenerative diseases [1,2,3,4,5,6]. The 26S proteasome contains a multi-catalytic enzyme complex mediating protein degradation, the 20S core, which is found predominantly in 2 forms. The constitutive proteasome is expressed ubiquitously throughout the body and is responsible for protein degradation in tissues, including heart, kidney, and liver. In the constitutive proteasome, proteolytic activities are encoded in the β5, β1, and β2 subunits [7,8,9]. The immunoproteasome is expressed mostly in immune cells and contains low molecular mass polypeptide (LMP 7, LMP2), and multi-catalytic endopeptidase complex-like 1 (MECL-1) instead of β5, β1, and β2, respectively, as catalytic subunits (Figure 1). Inhibition of both types of proteasome results in antitumor activity via induction of apoptosis, while selective inhibition of the immunoproteasome results in cytokine blockade in vitro and anti-inflammatory activities in in vivo models [10,11,12,13].

Dual proteasome inhibition has been clinically validated through regulatory approval of 3 compounds, all with equipotency for β5 and LMP7, for use in the treatment of B-cell neoplasms, such as multiple myeloma and mantle cell lymphoma [14,15,16,17]. These compounds comprise 2 distinct chemical classes: the reversible boronic acid derivatives, bortezomib (VELCADE^®^) and ixazomib (NINLARO^®^), and the irreversible covalent tetrapeptide epoxyketone-based compound, carfilzomib (KYPROLIS^®^). These three proteasome inhibitors (PIs) are currently approved for the treatment of multiple myeloma (MM) or mantle-cell lymphoma (MCL) [18,19]. Another epoxyketone-based inhibitor, KZR-616, which selectively targets the immunoproteasome, is currently being evaluated in Phase 2 clinical trials in patients with autoimmune disorders, including Lupus Nephritis (LN), Dermatomyositis (DM), and Polymyositis (PM) (Figure 2, Table 1) [20].

Covalent drugs have traditionally been considered to be less favorable clinically due to the potential for off-target reactivity [21,22]. However, the success of the approved covalent inhibitors has led to a resurgence in the study and development of this class of small molecule agents. A covalent mechanism involves the irreversible binding of the inhibitor to the target resulting in potent inhibition in a biochemical, cellular, and in vivo setting. Two factors determine in vivo covalent occupancy: a physiologically relevant k_inact_/K_i_ and an unbound AUC derived from a PK profile [23]. Boronic acid derivatives and epoxyketone-based proteasome inhibitors interact with the catalytic threonine residue within target catalytic subunits to exert their inhibitory activity. Boronic acid inhibitors form a slowly reversible tetrahedral intermediate between the boronate warhead of the molecule and the γ-OH side chain of the N-terminal threonine. In contrast, the mechanism of action of epoxyketone-based inhibitors involves nucleophilic attack of the γ-OH side chain of the N-terminal threonine residue on the epoxide carbonyl, followed by a second nucleophilic attack by the free α-NH_3_ group of threonine on the epoxide ring, to form an irreversible, dual covalent adduct with the proteasome active site [24,25,26].

Dual proteasome inhibition is thought to mediate an anti-tumor response through direct induction of apoptosis. This is thought to occur via effects on multiple pathways, including blockade of the transcription factor NF-kB, alteration in productive cell cycle control, and the accumulation of unfolded and misfolded proteins, which trigger the unfolded protein response (UPR) [27]. Interestingly, selective inhibition of either the constitutive proteasome or the immunoproteasome in myeloma cells failed to induce apoptosis. In contrast to affecting cell survival, selective inhibition of the immunoproteasome with agents, such as KZR-616, mediate an anti-inflammatory response without affecting cell survival. The effects of immunoproteasome inhibition include reduced expression of cytokines from activated immune cells, blockade of T helper 1 and 17 cell differentiation, and reduced fitness of plasma cells [28]. In mouse models of several autoimmune disorders, including rheumatoid arthritis and systemic lupus erythematous, KZR-616 blocked disease progression at well-tolerated doses without apparent signs of immunosuppression [29].

Atomic level molecular structures of proteasomes with 3 epoxyketone-based inhibitors, carfilzomib, oprozomib, and ONX 0914 (an immunoproteasome selective inhibitor), have been solved and reveal distinct binding features conveying insights on potency and selectivity [30]. Evidence from both x-ray crystallography and electron cyro-microscopy of the human constitutive 20S proteasome with and without the dual inhibitors carfilzomib or oprozomib bound at resolutions as low as 1.9 angstroms indicates that the S3 binding pockets play a pivotal role in both agents selectivity for β5 [26,31]. These studies also demonstrate 2 possible dual covalent interaction modes between the epoxyketone warhead with the catalytic N-terminal threonine, a 1,4 morphiline, or 1,4 oxazapene adduct. Structural comparison with the mouse immunoproteasome and constitutive proteasome revealed differences at the S1 portion just proximal to the catalytic threonine within the chymotrypsin-like subunits [32]. In LMP7, the side chain of the methionine at position 45 is stabilized by the aliphatic side chain of a conserved glutamine at position 53, resulting in a larger S1 pocket than in β5. The P1 phenylalanine of ONX 0914 is postulated to create steric hindrance in the smaller S1 pocket of β5, thus reducing potency for this subunit relative to LMP7. Oprozomib, which also contains a P1 tyrosine, contains a methoxy group at the P3 position (versus alanine in ONX 0914), which is suitable for making van der Waals interactions with the hydrophobic S3 pocket of both chymotrypsin-like subunits, thereby explaining dual targeting nature of this compound [30]. KZR-616 has similar binding modes as ONX 014 with P1 cyclopentene replacing phenylalanine and an R-hydroxyl group substitution at the β position of the P2 methyltyrosine side chain, resulting in hydrogen bonding with serine at position 21 [20].

While the structure, and function of the proteasome and the mechanisms of action of various proteasome inhibitors and their pharmacology has been reviewed extensively in the literature [33,34,35,36], there is no review on the metabolism of clinically applied proteasome inhibitors so far. In this review, we summarize the metabolism, pharmacokinetics, and pharmacodynamics of the three clinically-approved proteasome inhibitors and compare those profiles across chemical classes. Furthermore, we will discuss the unique features of metabolism of peptide epoxyketone-based proteasome inhibitors that potentially enable use of this chemical class in the treatment of chronic conditions.

## 2. Pharmacokinetic, Pharmacodynamics, and Metabolism of Boronic Acid-Based Dual Proteasome Inhibitors

Bortezomib, a dipeptide boronic acid derivative, was the first proteasome inhibitor to receive regulatory approval and is a mainstay in the treatment of multiple myeloma and other plasma cell malignancies [14]. Bortezomib is formulated as a mannitol ester and delivered to patients via intravenous (IV) or subcutaneous (SC) route, following one of two schedules of administrations consisting of 21-day cycles with dosing on Days 1, 4, 8, and 11 or Days 1 and 8. Exposure was equivalent with both routes of administration at 1.3 mg/m^2^, and no differences in anti-tumor activity were noted between SC- and IV-treated patients [35]. Clearance of bortezomib is rapid in patients; however, accumulation is seen upon repeated dosing. As a single agent, bortezomib induced a dose-dependent inhibition of 20S proteasome activity from 36%, 60%, 65%, to 74%, after 1-h treatment of at 0.40-, 1.04-, 1.20-, and 1.38-mg/m2 doses, respectively [37]. Proteasome inhibition was also similar with both routes of administration. The mean maximum inhibition (E_max_) was 57 to 63.7% for SC, and approximately 69% for IV administration. Time to E_max_ was longer (median of 120 min versus 5 min) for SC administration, likely due to longer t_max_ versus IV administration [36,38,39]. Maximum proteasome inhibition was measured 1 h after administration with complete recovery of proteasome activity within 72 h. The relationship between plasma concentration and proteasome inhibition was assessed at 1, 6, and 24 h over a 24-hour period. Recovery of proteasome activity was observed by 24 h, and the recovery was homogeneous. Interestingly, at 1 h post-dose, plasma concentration varied by ~800%, while proteasome inhibition varied only by ~70%, indicating a disconnection between plasma concentration and target inhibition [40,41]. It is noteworthy that, despite similar drug exposure and proteasome inhibition levels, the rates of peripheral neuropathy, the adverse event that represents the primary dose limiting toxicity, was reduced in the SC-treated patients versus. Those receiving IV administration of bortezomib likely reflect the different PK profiles [35].

The predominant pathway elimination of bortezomib is through oxidative de-boronation, mostly to a pair of diastereomeric carbinolamide metabolites [42]. In vitro metabolite identification (metID) studies revealed that the major phase 1 metabolic reactions are mediated by cytochrome P450 isomers 3A4 and 2C19, while phase II conjugation pathways do not appear to play any major role in bortezomib metabolism. The inactive deboronated metabolites (M1 and M2) then undergo a series of hydroxylations leading to their elimination. In animals and humans, bortezomib is extensively metabolized with more than 30 metabolites identified, including M1 and M2. Consistent with the in vitro studies, bortezomib is primarily metabolized in vivo via CYP450 enzymes (predominantly 3A4 and 2D6) and not phase II pathways, such as glucuronidation and sulfation. The major metabolites detected in rat bile and monkey fecal samples were M1 and M2 [43]. These same two metabolites, along with M4, were the major metabolites detected in human plasma, suggesting similar metabolic pathways for bortezomib metabolism in rodents, non-human primates, and humans (Figure 1). Bortezomib is a poor inhibitor of recombinant CYP450 isozymes 1A2, 2C9, 2C19, 2D6, and 3A4 with IC_50_ values of greater than 18 µM (~7 µg/mL) in human liver microsomes. These IC_50_ values are 60-fold higher than the C_max_ (89–120 ng/mL) observed in patients treated with labeled dose of 1.3 mg/m^2^. Therefore, it is unlikely that direct CYP450 inhibition is the mechanism underlying the apparent accumulation of bortezomib in humans following repeat dosing. However, in drug-drug interaction (DDI) studies, patients receiving bortezomib plus ketoconazole, a CYP3A4 inhibitor, revealed an increase in bortezomib exposure by 1.4-fold [44], further indicating the role of CYP3A4 in the metabolism of this drug.

### Ixazomib

Ixazomib is an orally bioavailable analog of bortezomib and is approved for the treatment of relapsed multiple myeloma [38]. As in bortezomib, ixazomib preferentially binds to and inhibits the chymotrypsin-like activities of the 20S proteasome (β5 and LMP7) with 10- and 1000-fold less potency for caspase-like and trypsin-like activities, respectively. In clinical studies using IV administration, proteasome inhibition in whole blood samples was rapid and dose-dependent and recovered within 24 h dosing [45]. In patients, ixazomib shows rapid absorption with T_max_ of approximately 1 h, low apparent clearance of ~1.9 L/h with a t_1/2_ of 9.5 days, and oral bioavailability of 58%. In a phase 1 study in relapsed/refractory lymphoma patients, plasma exposure increased dose proportionally from 0.5–3.11 mg/m^2^, which was also population PK analysis, and no apparent relationship between oral dose (0.2–10.6 mg) and plasma clearance [46]. At clinically relevant concentrations, no single CYP450 isozymes appear to play dominant role in the clearance of ixazomib using microsomes containing recombinantly expressed human CYP enzymes. However, at higher concentrations, ixazomib was metabolized by multiple CYP isozymes with contributions by CYP3A4 (42%), CYP1A2 (26%), and CYP2B6 (16%). Unlike bortezomib, DDI studies have shown no significant effects of strong inhibitors of CYP3A4 on the exposure to ixazomib. Non-CYP pathways seems to be major contributor for the metabolism of ixazomib [39]. The proposed metabolic pathway of ixazomib is shown in Figure 2, based on the phase 1 study using ^14^C-ixazomib in patients with advanced solid tumors or lymphoma. Metabolite profiles were similar in plasma, urine, and feces from the four patients after administration of a single 4.1-mg oral dose of [^14^C]-ixazomib with total radioactivity (TRA) of ~500 nCi. All metabolites identified were de-boronated. Ixazomib (54.2% of plasma TRA) and metabolites M1 (18.9%), M2 (7.91%), M3 (10.6%) were the primary components in AUC_0–816h_ time-proportional pooled plasma. Hydrolytic metabolism in conjunction with oxidative de-boronation are the major metabolic pathways for ixazomib [47].

## 3. Pharmacokinetics, Pharmacodynamics, and Metabolism of Epoxide-Based Proteasome Inhibitors

### 3.1. Carfilzomib

Carfilzomib is a peptide epoxyketone-based proteasome inhibitor originally derived from the natural product epoxmicin [18]. It is the only approved agent with a reactive epoxide pharmacophore, a feature previously thought unsuitable for drug development [48]. Carfilzomib received initial FDA approval in relapsed and refractory myeloma in 2012 and is currently approved for use with IV administration (predominantly as a 30-min infusion) once or twice weekly at doses ranging from 27 to 70 mg/m^2^. It is unclear why a broader range of therapeutically active and tolerated doses of carfilzomib are able to be utilized relative to the approved boronate inhibitors, but carfilzomib has been demonstrated to have fewer off-target effects relative to bortezomib [15]. In addition, in a 2 head-to-head study compared to bortezomib, rates of peripheral neuropathy were >60% lower in carfilzomib-treated patients [49,50]. Carfilzomib cleared rapidly with t_1/2_ values of <30 min in humans and systemic clearance rates higher than hepatic blood flow, indicating extra-hepatic clearance mechanisms [51]. Exposure to carfilzomib, both maximum and total, increased with the dose but in a manner that was not dose-proportional [52]. Although an irreversible inhibitor, clearance in animals was unaffected with prior administration of another covalent proteasome inhibitor, suggesting that target binding does not mediate the rapid clearance seen in animals and humans [53]. When administrated as a 30-min infusion to rats at 8 mg/kg, the steady state concentration (C_ss_) was 28-fold lower than the C_max_ in the bolus group at the same dose, while the total exposure (AUC) and proteasome inhibition in blood and tissues were equivalent. These results demonstrate that the level of target inhibition achieved in vivo is correlated with the total dose but not C_max_ [54].

IV administration of carfilzomib resulted in suppression of proteasome CT-L activity in patients when measured in blood one hour after the first dose. Inhibition of proteasome CT-L activity was comparable in whole blood (predominantly constitutive proteasome) and PBMCs (immunoproteasome), indicating that carfilzomib acts as a dual proteasome inhibitor. Using an active site binding assay called ProCISE [44], first dose inhibition of the β5 and LMP7 subunits of the constitutive proteasome and immunoproteasome, respectively, ranged from 67 to 74% and 77–80%, respectively, at 15 mg/m^2^ and 20 mg/m^2^. Inhibition of the LMP2 and MECL-1 subunits of the immunoproteasome ranged from 22 to 33 and 31 to 46% at 20 mg/m^2^, respectively. Less than 18% inhibition of the β1 or β2 was observed in whole blood samples at doses as high as 45 mg/m^2^. Measured occupancy of the other subunits increased at 45 mg/m^2^ dose with most prominent effect on MECL1 and LMP2. Proteasome inhibition was maintained for ≥ 48 h following the first dose of carfilzomib for each week of dosing. Near-complete recovery of proteasome activity was observed in PBMC between cycles.

Rapid metabolism in vitro by liver microsomes, in the presence of cofactor NADPH, indicated that carfilzomib is a substrate of CYP450 enzymes. However, in animals and humans in vivo, the predominant plasma and urine metabolites are inactive hydrolysis products, including the diol, and less than 1% of the dose was excreted intact (Figure 3) [54,55,56]. Additional and non-traditional studies were carried out in vitro and in vivo to reveal the roles of peptide cleavage and epoxide hydrolysis in the metabolism of carfilzomib [55]. In rat blood and tissue homogenates from liver, lung, and kidney, carfilzomib rapidly disappeared with the formation of 3 metabolites, the inactive diol metabolite M16, and the peptide cleavage products M14 and M15. Pooled plasma and urine samples from a phase 1 trial were used to determine the metabolite fate of carfilzomib following administration. The major metabolites M14, M15, and M16 were derived from peptidase cleavage and epoxide hydrolysis of carfilzomib, and CYP-mediated metabolites were detected only at very low levels [54]. M14 and M15 may have been derived from either carfilzomib or M16 (Figure 3). In cultured hepatocytes, the diol was the predominant metabolite further supporting the importance of epoxide hydrolysis. Additional studies using recombinant enzymes revealed that microsomal epoxide hydrolase (mEH), and not soluble epoxide hydrolase (sEH), is responsible for the formation of M16 (unpublished results). These studies revealed that carfilzomib metabolism is unique and not mediated by standard CYP450 or phase II enzymes. In support of this, there is no apparent alteration in exposure to carfilzomib in patients with impaired hepatic function [57].

### 3.2. Oprozomib

As in carfilzomib, oprozomib is a peptide epoxyketone-based proteasome inhibitor and is currently in clinical trials [58]. Oprozomib is a result of a chemistry effort to discover orally bioavailable analogues of carfilzomib which would have potential for improved flexibility in dosing and patient convenience over intravenously administrated agents. The effort focused on structure and activity relationship of short peptide portion of carfilzomib since di- and tripeptides can cross intestinal epithelial barriers, while tetrapeptides are generally not orally bioavailable. The tripeptide oprozomib demonstrated an equivalent potency for ß5 and LMP7, displayed selectivity for the chymotrypsin-like (CT-L) subunits over trypsin-like (T-L), and (caspase-like) C-L activities of the proteasome, and promoted an equivalent antitumor response to carfilzomib in mouse models of human tumors. Oprozomib showed improved solubility, metabolic stability and displayed a moderate oral bioavailability with F% of 17, 21, and 39 in mice, rats, and dogs, respectively. Oral bioactivity measured by PK and PD was found to be comparable [59]. Oprozomib has shown clinical activity in the patients with hematologic malignancies. The maximum tolerated dose (MTD) of single agent oprozomib was 300 mg/day when administrated the first 2 days every 7 days of a 14-day cycle or 240 mg/day when administrated on the first 5 days of a 14-day cycle. Oprozomib was administrated in capsule and tablet forms in patients with multiple myeloma and as tablets in patients with Waldenström’s macroglobulinemia. Absorption was rapid with median t_max_ between 0.4 and 2 h, both tablets and capsules. Clearance was also rapid with a mean t_1/2_ ranging from 0.52 to 2.5 h [60]. Exposure to oprozomib increased dose proportionally in patients with multiple myeloma with a high inter-patient variability [61]. Potent proteasome inhibition was observed in whole blood 8 h after administration. At the maximum tolerated dose (MTD; 300 mg/day on the 2/7 schedule and 240 mg/day on the 5/14 schedule), proteasome inhibition was ≥70% 4 h post-dose [61].

Metabolic stability of oprozomib was conducted in human hepatocytes, and a predominant diol of oprozomib (PR-176) was formed from direct epoxide hydrolysis. Two metabolites at relatively low levels were observed from peptide bond cleavage: one with *m*/*z* at 245.0587 (bond 2 cleavage), and the other with *m*/*z* at 224.1276 (diol od PR-025, bond 1 cleavage). Other trace amount metabolites from hydroxylation and/or de-methylation, direct glutathione (GSH) conjugation, and the combination of oxidation and epoxide hydrolysis or GSH conjugation were also observed. Quantification using synthesized standards revealed that ~55% of total oprozomib was recovered as the diol, ~7% recovered as the diol of PR-025, and 20% remained as oprozomib. These results suggested that epoxide hydrolysis is the major metabolic pathway for oprozomib in human hepatocytes. Metabolic stability of oprozomib in the presence and absence of NADPH was evaluated in human liver microsomes (HLM). In pooled HLMs, the presence of NADPH significantly increased the intrinsic clearance of oprozomib by 3.5-fold compared with that in the absence of NADPH (70.7 ± 2.9 versus 20.2 ± 1.8 µL/min/mg protein), indicating a significant role of oxidation pathways over epoxide hydrolysis in HLMs, which is in contrast with the result observed in human hepatocytes. Recombinant enzyme studies demonstrated that epoxide hydrolysis activity is mediated by mEH, rather than sEH, and the in vitro metabolism is consistent with the findings from in vivo samples (Figure 4) [62]. There is no published metID from patient samples.

### 3.3. KZR-616

Use of the tool compound ONX 0914 revealed that selective inhibition of the immunoproteasome results in broad and potent anti-inflammatory activity in preclinical models of multiple autoimmune disorders [10,63,64]. KZR-616 is a tripeptide ketoepoxide-based selective inhibitor of the human immunoproteasome that was derived from a medicinal chemistry effort that involved the synthesis of over 400 tripeptide epoxyketones. The design of these inhibitors utilized a human homology model evolved of the murine crystal structure of the ONX 0914/proteasome isoform complexes. The model was developed to elucidate per subunit the optimal recognition of nonprime selectivity pockets (S1, S2, S3) by the amino acid side chains/features (P1, P2, and P3, respectively (Figure 3)) [20]. Profiling of selective inhibitors in PBMCs stimulated to produce cytokines revealed that the inhibition of LMP7 alone was insufficient to inhibit broad cytokine expression and translated to minimal activity in a mouse model of inflammatory arthritis. A combination of LMP7 and LMP2 and/or MECL-1 inhibitors offered an optimal cytokine inhibition profile in vitro, which correlated with in vivo efficacy. It is noteworthy that KZR-616 achieved activity in the mouse arthritis model via complete LMP7 inhibition and ~40% LMP2 inhibition, suggesting that only partial inhibition of a secondary immunoproteasome subunit is sufficient for a multi-cytokine inhibitory effect. The discovery effort also improved the physicochemical properties of KZR-616 for later clinical use [20].

In healthy volunteers, subcutaneous (SC) administration of KZR-616 demonstrated rapid absorption with a T_max_ < 1 h and rapid clearance with a t_1/2_ < 4 h. Exposure was linear across a 10-fold dose range, and no accumulation was observed with repeated weekly dosing. The bioavailability of SC administration was 70–100% and low inter-subject variability was noted in total exposure (the major determinant for target inhibition). Across doses of 7.5 to 75 mg, mean inhibition of PBMC chymotrypsin-like (CT-L) activity (predominantly immunoproteasome) measured 4 h post-dose ranged from 56 to 93%. Mean inhibition of whole blood CT-L activity (predominantly constitutive proteasome) ranged from 8 to 43% across a tested dose range of 7.5 to 60 mg, demonstrating that KZR-616 selectively inhibits the immunoproteasome relative to the constitutive proteasome (Figure 4). Following repeat-dose SC administration, there was no apparent accumulation of immunoproteasome inhibition, although cumulative constitutive proteasome inhibition in whole blood was observed, likely due to the irreversible mechanism of KZR-616 inhibition and the inability of enucleated red blood cells (the major cellular source of constitutive proteasome) to produce new proteasomes. PK and PD correlation analysis reveals that immunoproteasome inhibition is related to total (R^2^ = 0.712) rather than maximum (R^2^ = 0.464)) exposure (Figure 5) [65]. As in carfilzomib, and oprozomib, KZR-616 was not stable when incubated in HLMs in the absence of NADPH, indicating non-CYP enzymes were involved in its clearance. CYP phenotyping demonstrated that CYP3A4/5 was the major enzyme involved in the microsomal metabolism of KZR-616, with CYP2C8 playing a minor role in its metabolism. Formation of the KZR-616 diol, KZR-59587, was not affected by the presence of CYP inhibitors, demonstrating that other enzymes, such as epoxide hydrolases, play important roles in diol formation. Metabolic profiling using hepatocyte suspensions from rats, monkeys, and humans revealed KZR-59587, was the most predominant metabolite, and no metabolites unique to humans were observed [66]. In pooled plasma samples from 4 patients with systemic lupus erythematous (SLE) receiving a KZR-616 dose of 75 mg, the diol was the major metabolite, and other metabolites from oxidation, hydrolysis + dehydrogenation, oxidation + dehydrogenation double oxidation were less than 1% of the total peak area of all components (Table 2).

To further evaluate the role of epoxide hydrolase in the metabolism of KZR-616, two analogs were selected for PK and PD studies in cynomolgus monkey to compare with KZR-616. Relative to KZR-616, KZR-59240 demonstrated increased stability, while KZR-59177 demonstrated reduced stability when cultured in liver microsome suspensions in the absence of NADPH. Following single subcutaneous (SC) administration to monkeys at 3 mg/kg of these 3 compounds AUC values were the highest with KZR-59240, followed by KZR-616 and KZR-59177. The PK of both parent and diol for each compound were consistent with the stability results in MLM in the absence of NADPH, in which the CYP enzymes were not activated (Table 3), and, as predicted, the diol derivatives were determined to be the major metabolites for all compounds. The PK and PD of KZR-616 was assessed using 2 different drug product formulations, one containing 10% (*w*/*v*) PS-80, and the other 2% (*w*/*v*) trehalose. PK analysis of this study showed a 2-fold difference in C_max_ but equivalent total exposure (AUC) for KZR-616 in both formulations. Since the PD was roughly equivalent between the 2 different formulations, these data support the hypothesis derived from studies with carfilzomib that target inhibition is an effect of total, but not maximum, drug exposure.

Similar to carfilzomib, mEH-mediated diol formation of KZR-616 (Figure 5) and was not saturable in vitro. However, unlike both carfilzomib and oprozomib, no peptide cleavage products were identified, suggesting that diol formation is the sole metabolic pathway of KZR-616 in humans. KZR-616 is currently being evaluated in Phase 2 clinical trial in patients with lupus nephritis (LN) (MISSION Study; NCT03393013), as well as polymyositis and dermatomyositis. (PRESIDIO Study; NCT04033926).

## 4. Discussion and Conclusions

Small molecule targeting of the catalytic subunits of the proteasome has been a fruitful area of drug discovery and development. Currently approved proteasome inhibitors comprise 2 chemical classes with distinct pathways of metabolism and clearance (Table 1). Despite these differences, boronic acid-based and peptide epoxyketone-based inhibitors all show rapid clearance in animals and humans. However, the pharmacodynamic profile of boronic acid-based and peptide epoxyketone-based inhibitors were different in patients. At doses of carfilzomib routinely used in the clinic (i.e., ≥15 mg/m^2^), average inhibition of β5 and LMP7 was greater than 83% [55]. This level of inhibition is higher than the 65–70% measured in bortezomib-treated patients [37,40,67]. There was little to no recovery of proteasome activity in whole blood following the administration of carfilzomib which is consistent with preclinical results in rodents, though this likely does not reflect effects in tissues [18]. Interestingly, the recovery of activity in PBMCs was slower than predicted-based on animal data. Minimal recovery of proteasome activity was seen 24 h after the first dose, and incomplete recovery was observed after the 5 day non-dosing period during the first week of treatment. In contrast, 50% recovery of activity was observed 12 h after administration of bortezomib, with full recovery achieved 36 h after dosing [68]. The differences in proteasome recovery between the two drugs may be due to the irreversible bonding of carfilzomib in contrast to the reversible binding of bortezomib [55], rather than differences in exposure parameters. While the recovery of proteasome activity in carfilzomib-treated subjects relies completely on new proteasome regeneration, recovery in bortezomib-treated subjects involves both release from bound proteasome active sites and new proteasome generation [52]. Metabolism of bortezomib and ixazomib are mediated by CYP450s, and possibly additional enzymes, while peptide epoxyketone inhibitors, such as carfilzomib and oprozomib, are cleared via epoxide hydrolase activity and peptide hydrolysis. Epoxide hydrolases are a ubiquitously expressed family of enzymes found in mammals [69,70,71,72] and are the primary pathway for the detoxification of compounds containing an epoxide residue [73]. There are 2 major isoforms, mEH, encoded by the EPHX1 gene and localized predominantly in the endoplasmic reticulum [74], and sEH, encoded by the EPHX2 gene and confined mainly to cytoplasm [75]. EPHX1 is well recognized as one of the main enzymes detoxifying xenobiotic epoxides. Owning to the high chemical reactivity of these compounds, EPHX1 was considered to be protective against mutagenic and carcinogenic initiation. Consequently, EPHX1 inhibition is thought to be rather deleterious, since it increases the toxicity of xenobiotics, as well as the risk of cancer and inflammatory diseases. On the other hand, EPHX2 was shown to orchestrate a variety of physiological functions, by mediating the formation of cytotoxic dihydrodiols at the expense of rather cytoprotective epoxides of fatty acids [76]. Consequently, EPHX2 inhibition emerged as a promising therapy to treat several diseases, including neuropathic pain, diabetic peripheral neuropathy, and averting cytokine storms in COVID-19 [77,78,79]. However, the reality seems to be more complex. More recently, EPHX1 was shown to play a leading role in the hydrolysis of different fatty acid epoxides [80]. The approval of carfilzomib as the first small molecule containing a reactive epoxide opens up the possibility of additional novel agents containing this reactive pharmacophore. However, additional or alternative studies for metabolism, including the use of tissue homogenates, are required to understand the metabolic fate of these compounds. Interestingly, carfilzomib, oprozomib, and KZR-616 were all found to be substrates of mEH but not sEH, although their peptide sequence and size are different. The active site of mEH harbors two conserved tyrosine residues which may contribute to substrate specificity and orientation of substrates within the active site [81]. The epoxide ring in these three proteasome inhibitors can fit into the active pocket of mEH with a right position suitable for hydrolysis to form the corresponding inactive diols [82]. CYPs play minimal role for the metabolism of proteasome inhibitors containing epoxide moiety. Metabolic findings using liver microsomes should be interpreted with caution since the microsomal system may be biased toward P450-mediated metabolism and may underestimate non-P450 metabolism. It is also noteworthy that KZR-616 was not subject to peptide hydrolysis as a means of metabolism. This singular and non-saturable metabolic pathway may help explain the consistent PK noted in humans thus far and may prove to be a salient feature for its use as a long-term chronic therapy (Table 1).

## Data Availability

Not applicable.

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
