# Peer review of "Proteasome Inhibitors and Their Pharmacokinetics, Pharmacodynamics, and Metabolism"

_ijms, 2021, doi:10.3390/ijms222111595_

Round 1

Reviewer 1 Report

Here, in this comprehensive study the authors provide the comparative analysis of proteasome inhibitors in context of its metabolism, pharmacokinetics and pharmacodynamics. This review  is of significant interest for the researchers studying the involvement of proteasomes in molecular mechanisms related to oncogenesis. Furthermore it may be of significant interest to the chemists who develops novel effective proteasome inhibitors.

Author Response

We would like to thank Reviewer 1 for his positive comments and inputs.

Reviewer 2 Report

In the present review, J. Wang and co-authors deals the pharmacokinetics, pharmacodynamics and metabolism of some proteasome inhibitors. However, the review appear excessively concise and poor of content.

It needs of major revisions.

  • The mechanism of action of different inhibitors should be described and discussed
  • The main difference in the structure and action between standard and constitutive proteasome should be discussed before the section 3.3 as well as the structural features that renders selective the inhibitor KZR-616
  • In the different schemes, a caption could help to understand the different metabolic pathways.
  • Table 1 and Table 4 could be merged
  • In the table 1 the column “indication” is little informative, please remove or furnish more details.
  • Concerning therapeutic indications, the authors should include more details in the main text on in a box section.
  • Some important very recently review on proteasome lack in the bibliography and should be cited, for example:

Pharmacol Ther. 2020 Sep;213:107579. doi: 10.1016/j.pharmthera.2020.107579. Epub 2020 May 19.

Biochemistry (Mosc . 2019 Jan;84(Suppl 1):S159-S192. doi: 10.1134/S0006297919140104.

Author Response

In the present review, J. Wang and co-authors deals the pharmacokinetics, pharmacodynamics and metabolism of some proteasome inhibitors. However, the review appear excessively concise and poor of content.

It needs of major revisions.

A: We would like to thank Reviewer 2 for his comments and suggestions.

  • The mechanism of action of different inhibitors should be described and discussed

A: We added detailed mechanism of action of two distinctive proteasome inhibitors (line 66 to 79).

  • The main difference in the structure and action between standard and constitutive proteasome should be discussed before the section 3.3 as well as the structural features that renders selective the inhibitor KZR-616

A: We added detailed discussion before section 3.3 regarding the main difference in the structure and action between constitutive and immunoproteasome and we also discussed the structural features that renders selective inhibitor of KZR-616 based on proteasome: inhibitor crystal structure and modeling (line 80 to 101).

  • In the different schemes, a caption could help to understand the different metabolic pathways.

A: Captions have been added for the different metabolic pathways

  • Table 1 and Table 4 could be merged

A: Table 1 and Table 4 were merged to become Table 1

  • In the table 1 the column “indication” is little informative, please remove or furnish more details.

A: The column “indication” in Table 1 was removed

  • Concerning therapeutic indications, the authors should include more details in the main text on in a box section.
  • A: More details have been added regarding therapeutic indications
  • Some important very recently review on proteasome lack in the bibliography and should be cited, for example:

Pharmacol Ther. 2020 Sep;213:107579. doi: 10.1016/j.pharmthera.2020.107579. Epub 2020 May 19.

Biochemistry (Mosc . 2019 Jan;84(Suppl 1):S159-S192. doi: 10.1134/S0006297919140104.

A: We have cited more references including the two mentioned above.

Round 2

Reviewer 2 Report

In this revised version the work is appropriate for the publication